# COVID-19 IgG seropositivity and its determinants in occupational groups of varying infection risks in two Andean cities of Ecuador before mass vaccination

**Jose E. Leon-Rojas**[1,2]*, **Fernanda Arias-Erazo**[3,4], **Patricia Jiménez-Arias**[2,4,5], **Ricardo Recalde-Navarrete**[2,6], **Angel Guevara**[7], **Josefina Coloma**[8], **Miguel Martin**[1,2,3], **Irina Chis Ster**[9], **Philip Cooper**[3,9], **Natalia Romero-Sandoval**[2,3], on behalf of the Seroprevalence ECU-Group[¶]

1 Departamento de Pediatría, Obstetricia y Ginecología y Medicina Preventiva, Universitat Autonoma de Barcelona (UAB), Barcelona, Spain, 2 Research Network Grups de Recerca d'Amèrica i Àfrica Llatines (GRAAL), Universidad Internacional del Ecuador, Quito, Ecuador, 3 School of Medicine, Universidad Internacional del Ecuador, Quito, Ecuador, 4 Grupo de Investigación en Sanidad Animal y Humana (GISAH) ESPE, Quito, Ecuador, 5 Departamento de Ciencias de la Vida y de la Agricultura, Universidad de las Fuerzas Armadas ESPE, Sangolquí, Ecuador, 6 Medical School, Universidad Tecnica de Ambato, Ambato, Ecuador, 7 Universidad Central del Ecuador, Quito, Ecuador, 8 School of Public Health, University of California, Berkeley, Berkeley, CA, United States of America, 9 Institute of Infection and Immunity, St George's University of London, London, United Kingdom

¶ Membership of the Seroprevalence ECU-Group is provided in the Acknowledgments.
* JoseEduardo.Leon@autonoma.cat

## Abstract

### Background

The COVID-19 pandemic has caused over 68.7 million infections and 1.35 million deaths in South America. There are limited data on SARS-CoV-2 seropositivity and its determinants from Andean countries prior to mass vaccinations against COVID-19.

### Objective

To estimate SARS-CoV-2 seropositivity and its determinants before vaccination in occupational groups of adults presumed to have different levels of exposure and associations with potential symptomatology.

### Methods

We measured seropositivity of anti-SARS-CoV-2 IgG antibodies in a cross-sectional study of vaccine-naïve adults aged 18 years and older, recruited within three occupational risk groups (defined as low [LR], moderate [MR], and high [HR]) between January and September 2021 in two Andean cities in Ecuador. Associations with risk factors were estimated using logistic regression.

### Results

In a sample of 882 adults, IgG seropositivity for the three different occupational risk groups was 39.9% (CI 95% 35.3–44.6), 74.6% (CI 95% 66.4–81.4), and 39.0% (CI 95% 34.0–44.4)

**Data Availability Statement:** The full simplified database, supporting our results, can be accessed openly through: https://bit.ly/4aa1Fdf.

**Funding:** This study was funded by the following grants: Universidad Internacional del Ecuador, Grants No. UIDE-DGIP-MAT-PROY-20-003; 21-007; 21-008 awarded to JELR, NRS, and PC. Corporación Ecuatoriana para el Desarrollo de la Investigación y Academia (CEDIA) Grant Number: CEPRA XVI-2022-17 ANTICUERPOS IGG IGA fund awarded to JELR, RR, PJ, and NRS. The funders had no role in study design, data collection and analysis, decision to publish, or preparation of the manuscript.

**Competing interests:** The authors have declared that no competing interests exist.

for the HR, MR, and LR groups, respectively. History of an illness with loss of taste and/or smell was significantly associated with seropositivity in all occupational groups, with adjusted ORs of 14.31 (95%CI, 5.83–35.12; p<0.001), 14.34 (95%CI 3.01–68.42; p<0.001), and 8.79 (95%CI 2.69–28.72; p<0.001), for the HR, MR, and LR groups, respectively; while fever was significant for the LR group with an adjusted OR of 1.24 (95%CI, 1.11–4.57; p = 0.025) and myalgia for the HR group with an adjusted OR of 2.07 (95%CI, 1.13–3.81; p = 0.019).

## Conclusion

Notable proportions of seropositivity were seen in all occupational groups between January and September 2021 prior to mass vaccination. Loss of taste and/or smell was strongly associated with presence of anti-SARS-CoV-2 IgG antibodies irrespective of presumed occupational exposure risk.

## Introduction

COVID-19, caused by the severe acute respiratory syndrome coronavirus 2 (SARS-CoV-2), emerged as a global pandemic estimated at 68.7 million infections and 1.35 million deaths in South America up to February 2024 [1]. Infection with SARS-CoV-2 generally leads to a measurable specific IgG response directed towards the viral spike glycoprotein and nucleocapsid protein within 3 weeks of symptom onset and persists for several months [2].

Prior to the introduction of mass vaccination campaigns against COVID-19, the detection of specific antibodies to SARS-CoV-2 using standard serological assays has been used to estimate the proportion exposed [3, 4] and provided a measure of spread of the infection within a population at a point in time [5, 6]. Numerous studies have reported SARS-CoV-2 seropositivity in different populations worldwide prior to the implementation of vaccination campaigns [7–19]. However, there are relatively few studies from the South American region assessing pre-vaccination seropositivity in occupational risk groups.

There is a considerable variability in the estimates associated with seropositivity for SARS-CoV-2 prior to COVID-19 vaccination. A meta-analysis of 965 SARS-CoV-2 seropositivity studies from 104 countries worldwide up to September 2021 estimated an overall seropositivity of 59.2% (95%CI 56.1%-62.2%) [20]. Another meta-analysis of 88 studies from 34 countries, with sampling dates up to November 2020, estimated a pooled seropositivity of 8% (95% CI 6–10%) in the Americas that was highest in Colombia (29%, 95% CI 23–31%) [2]. Analysis of nine seropositivity studies done in South America between April and September 2020 estimated a pooled seropositivity of 33.6% (95% CI 28.6–38.5%) [21], with a particularly high seropositivity of 73% observed in indigenous populations [10].

Factors associated with SARS-CoV-2 seropositivity have been widely studied [9, 10, 22–29] showing associations with a range of factors including symptoms, occupational exposures (with greater risks observed among healthcare workers, students, prison workers, cleaning staff, and highly mobile workers), and ethnicity [7, 9, 10, 22–31].

There are limited published information on pre-vaccination COVID-19 seropositivity in occupational groups of varying infection risks and associated factors in Ecuador [12–19]. In the present study, we estimated seropositivity for the presence of anti-SARS-CoV-2 IgG antibodies among COVID-19 vaccine-naïve adults, with varying levels of exposure to infection

(defined a priori as low, medium, and high based on presumed occupational risk) and associated factors, in two Ecuadorian Andean cities.

## Materials and methods

### Study design and participants

We conducted a cross-sectional study of vaccine-naïve adults between 13[th] January and 27[th] September 2021 to estimate seropositivity for anti-SARS-CoV-2 IgG antibodies in plasma. Participants were recruited from 3 occupational groups–according to perceived occupational risk of exposure into High Risk (HR), Moderate Risk (MR), and Low Risk (LR)—in two Andean cities of Ecuador; all participants signed a written informed consent. This study represents the analysis of baseline samples from a cohort study whose main objective was to analyse changes in antibody levels over a period one year following vaccination with several COVID-19 vaccines. The study protocol is described in detail elsewhere [32]. The HR group included participants in a health care setting at a major public hospital in Quito (city at altitude of 2,800 m with 2.9 million inhabitants in northern Andean region), including health care workers and support personnel working in the emergency service, intensive care units (ICUs), and hospitalization areas. The (MR) group included individuals whose occupations were highly mobile and who were unlikely to be careful with personal protection such as merchants, farmers or those working in markets in Huachi Grande, a rural parish in Ambato (city at altitude of 2,600 m with 500,000 inhabitants) in central Andean region. The (LR) sample aimed at representing people most likely to isolate such as teachers, administrative staff, and students at a private non-profit university in Quito. Exclusions were made based on pregnancy/breastfeeding, religious beliefs that prevented sample collection or inability to provide informed consent. Recruitment began with the HR group in January 2021 followed by LR and MR groups in July 2021.

### Questionnaire

Observations were collected by trained personnel using the ISARIC-WHO Global COVID-19 Clinical Platform Questionnaire (subsections 1b, 1f, and 3a) which included age, sex, ethnic group (categorized as mixed or non-mixed), self-reported history of testing for COVID-19 (including PCR and antibody testing), COVID-19 symptoms, and time between last COVID-19 diagnosis and sample collection. The full questionnaire can be found at S1 File. Additionally, we selected five symptoms (cough, fever, myalgia, headache, loss of taste and/or smell) to be analysed separately against seropositivity; these symptoms were chosen based on their frequency of presentation in COVID-19 patients [33, 34].

### Measurement of anti-SARS-CoV-2 IgG

Blood samples were collected into EDTA-containing Vacutainer 5mL tubes (Vacutainer, BD Biosciences). Plasma was separated by centrifugation and stored at -20˚C prior to analysis. The presence of IgG antibodies directed against the SARS-CoV-2 spike protein's Receptor Binding Domain (RBD) was evaluated in stored plasma using an ELISA assay adapted from Guevara et al., 2021; this test had a sensitivity, specificity, positive predictive value, and negative predictive value of 93.6%, 100%, 100%, and 95.4%, respectively [35].

### Statistical analysis

We planned to recruit up to 1,000 subjects. Descriptive statistics were presented stratified by the three exposure groups. Means, standard deviations, medians, inter-quartile ranges were

used to summarize continuous variables and frequencies and proportions were used to describe categorical variables.

A binary outcome indicating positivity/negativity following laboratory analyses has been investigated using logistic regression. Analyses were conducted on each of the samples from the three different groups. Univariate analyses explored direct associations between the outcome and each variable whilst multivariable logistic regressions were used to understand their adjusted effects. Associations were measured by odds ratios (ORs) and their corresponding uncertainties were quantified by the 95% confidence intervals (CIs). A p-value less than 0.05 indicated a statistically significant result, an OR greater (less) than 1 indicated a positive (negative) association between collected variables and outcome. SPSS Statistics (IBM) Version 29.0 was used for calculations.

### Ethics

The study protocol was approved by the Ethics Committee for COVID-19 Research of the Ecuadorian Ministry of Public Health (approval ID 020-2020-MSP-CGDES-2020-0172-O). Each participant voluntarily provided their consent by reading and signing an informed consent form.

## Results

889 participants were invited of whom 882 were recruited. Nine participants were excluded from the analysis because of inadequate blood samples. Of 882 participants included in the analysis, 422, 127, and 333 were in HR, MR, and LR groups, respectively. The mean age of each group was 40.5 years for HR, 40.1 for MR, and 20.5 for LR; most participants were of mestizo ethnicity (93.9%).

### Demographic, clinical, and social characteristics

Table 1 shows the socio-demographic and clinical characteristics for each of the three occupational risk groups for presumed exposure to SARS-CoV-2.

### Seropositivity for anti-SARS-CoV-2 IgG antibodies

The greatest seropositivity was estimated for the MR group (74.6%, 95% CI 66.4–81.4) while similar lower proportions were observed for HR (39.5%, 95% CI 34.9–44.4) and LR (39.0%, 95% CI 34.0–44.4) groups.

### Factors associated with anti-SARS-CoV-2 seropositivity

Univariate analyses (Table 2) showed that seropositivity was significantly associated with the participant's past medical history, specifically to a previous positive COVID-19 antibody or PCR test, but only in the HR and LR groups. With regards to past symptomatology (i.e., history of any of the 20 symptoms considered in our questionnaire), the presence of any symptom was associated with seropositivity only in the HR and LR groups, with crude ORs of 4.37 (95% CI, 1.91–10.01; p <0.001), and 1.94 (95%CI, 1.09–3.47; p 0.024), respectively. However, when examining individual COVID-19-associated symptoms, the symptom with the strongest association with seropositivity was loss of taste and/or smell with crude ORs of 51.02 (95% CI, 22.61–115.13; p<0.001), 15.65 (95% CI, 3.53–69.27; p<0.001), and 36.97 (95% CI, 12.95–105.61; p <0.001) for the HR, MR, and LR groups, respectively. Other flu-like symptoms also associated with seropositivity were cough, fever, and myalgia for the HR and LR groups, whereas headache was positively only associated in the HR and MR groups. Finally, there was

**Table 1. Sociodemographic and clinical characteristics of the three occupational risk groups.**

| Variable | High risk | | Moderate risk | | Low risk | |
|---|---|---|---|---|---|---|
| | n | % or range | n | % or range | n | % or range |
| **Sex** | | | | | | |
| Male | 142 | 33.6 | 49 | 38.6 | 209 | 62.8 |
| Female | 280 | 66.4 | 78 | 61.4 | 124 | 37.2 |
| **Ethnicity** | | | | | | |
| Mestizo | 392 | 92.9 | 116 | 91.3 | 320 | 96.1 |
| Non-mestizo | 30 | 7.1 | 11 | 8.7 | 13 | 3.9 |
| **Self-Reporting of PCR** | | | | | | |
| Positive | 162 | 38.4 | 10 | 7.9 | 45 | 13.5 |
| Negative | 145 | 34.3 | 6 | 4.7 | 92 | 27.6 |
| No Test | 115 | 27.3 | 111 | 87.4 | 196 | 58.9 |
| **Self-Reporting of Antibody Test** | | | | | | |
| Positive | 40 | 9.5 | 16 | 12.6 | 34 | 10.2 |
| Negative | 164 | 38.9 | 12 | 9.4 | 86 | 25.8 |
| No Test | 218 | 51.6 | 99 | 78.0 | 213 | 64.0 |
| **Time from COVID-19 Diagnosis–Median days (Q1-Q3)*** | 374 (372–376) | 13–408 | 132 (74–320) | 28–486 | 231 (110–370) | 0–506 |
| **SARS-CoV-2 IgG status** | | | | | | |
| Positive | 165 | 39.9 | 94 | 74.6 | 130 | 39.0 |
| Negative | 249 | 60.1 | 32 | 25.4 | 203 | 61.0 |
| Substandard Sample | 8 | | 1 | | 0 | |
| **Age (Mean–SD)** | 40.5 (9.9) | 20–69 | 40.1 (15.6) | 18–85 | 25.5 (8.0) | 18–55 |
| **Number of symptoms–Median (Q1-Q3)** | 4 (1–8) | 0–20 | 7 (3–12) | 0–20 | 2 (1–5) | 0–18 |
| **Number of classic symptoms–Median (Q1-Q3)** | 1 (0–3) | 0–6 | 2 (1–4) | 0–6 | 1 (0–2) | 0–6 |
| **Recruitment Dates Range in 2021** | 13 Jan—12 Feb | | 21 July– 27 Sept | | 3–21 July | |

HR: High Risk; LR: Low Risk; MR: Moderate Risk; PCR: Polymerase Chain Reaction; SD: Standard Deviation.

* Results based on 325 participants that reported a COVID-19 diagnosis

no association of ethnicity or sex with a seropositivity in all 3 occupational groups. In a sub-group analysis, we stratified HR participants into hospital personnel with direct patient contact (i.e., doctors, residents, medical students, nurses, nursing assistants, pharmacists, laboratory personnel, and nutritionists), and those personnel with minimal patient contact (i.e., administrative staff, kitchen, security guards, and maintenance). The first group consisted of 344 participants (67.4% women) with a mean age of 40.5 years (SD = 9.8), while the second group consisted of 78 participants (61.5% women) with a mean age of 40.9 years (SD = 10.3). We observed a greater risk of seropositivity among HR personnel with direct compared to minimal contact (OR 3.30, 95%CI 1.77–5.95, p<0.001).

In adjusted analyses, symptoms associated with seropositivity were myalgia in the HR group, fever in the LR group, and loss of taste and/or smell in HR (adjusted OR 30.8, 95%CI 13.20–71.86, p<0.001), MR (adj. OR 17.71, 95%CI 3.77–83.17, p<0.001), and LR (adj. OR 25.03, 95%CI 8.45–74.14, p<0.001) (Table 3). No other symptoms or sociodemographic factors were associated with seropositivity after statistical adjustment.

## Discussion

In the present study, conducted in two Ecuadorian Andean cities and in three groups with different perceived occupational risk of exposure (healthcare professionals, merchants, and

**Table 2. Univariate analysis of factors associated with seropositivity in the three occupational risk groups.**

| Variable | Category | HIGH RISK | | | | | MODERATE RISK | | | | | LOW RISK | | | | |
|---|---|---|---|---|---|---|---|---|---|---|---|---|---|---|---|---|
| | | Seronegative | Seropositive | OR | 95% CI | p-value | Seronegative | Seropositive | OR | 95% CI | p-value | Seronegative | Seropositive | OR | 95% CI | p-value |
| **Sociodemographic** | | | | | | | | | | | | | | | | |
| Sex | Male* | 90 (36.0%) | 49 (30.1%) | 1 | - | - | 14 (43.8%) | 34 (36.2%) | 1 | - | - | 127 (62.6%) | 82 (63.6%) | 1 | - | - |
| | Female | 160 (64.0%) | 114 (69.9%) | 1.31 | 0.85–1.99 | 0.21 | 18 (56.3%) | 60 (63.8%) | 1.37 | 0.61–3.11 | 0.446 | 76 (37.4%) | 47 (36.4%) | 0.96 | 0.61–1.51 | 0.853 |
| Ethnic Group | Non-Mestizo* | 18 (7.2%) | 11 (6.7%) | 1 | - | - | 4 (12.5%) | 7 (7.4%) | 1 | - | - | 8 (3.9%) | 5 (3.9%) | 1 | - | - |
| | Mestizo | 232 (92.8%) | 152 (93.3%) | 1.07 | 0.49–2.33 | 0.86 | 28 (87.5%) | 87 (92.6%) | 1.77 | 0.48–6.53 | 0.382 | 195 (96.1%) | 124 (96.1%) | 1.02 | 0.33–3.18 | 0.976 |
| **Past Medical History** | | | | | | | | | | | | | | | | |
| Self-Reporting Antibodies | No and No Test* | 238 (95.2%) | 135 (82.8%) | 1 | - | - | 29 (90.6%) | 81 (86.2%) | 1 | - | - | 197 (97.0%) | 101 (78.3%) | 1 | - | - |
| | Positive | 12 (4.8%) | 28 (17.2%) | 4.11 | 2.03–8.35 | <0.001¶ | 3 (9.4%) | 13 (13.8%) | 1.55 | 0.41–5.83 | 0.513 | 6 (3.0%) | 28 (21.7%) | 9.10 | 3.65–22.69 | <0.001¶ |
| Self-Reporting PCR | No and No Test* | 209 (83.6%) | 43 (26.4%) | 1 | - | - | 32 (100.0%) | 85 (90.4%) | # | # | # | 198 (97.5%) | 89 (69.0%) | 1 | - | - |
| | Positive | 41 (16.4%) | 120 (73.6%) | 14.22 | 8.77–23.06 | <0.001¶ | 0 (0.0%) | 9 (9.6%) | # | # | # | 5 (2.5%) | 40 (31.0%) | 17.8 | 6.79–46.61 | <0.001¶ |
| **History of Symptoms** | | | | | | | | | | | | | | | | |
| Presence of Symptoms | No* | 41 (16.4%) | 7 (4.3%) | 1 | - | - | 4 (12.5%) | 5 (5.3%) | 1 | - | - | 51 (25.1%) | 19 (14.7%) | 1 | - | - |
| | Yes | 209 (83.6%) | 156 (95.7%) | 4.37 | 1.91–10.01 | <0.001¶ | 28 (87.5%) | 89 (94.7%) | 2.54 | 0.64–10.12 | 0.173 | 152 (74.9%) | 110 (85.3%) | 1.94 | 1.09–3.47 | 0.024¶ |
| Cough | No* | 228 (91.2%) | 123 (75.5%) | 1 | - | - | 26 (81.3%) | 66 (70.2%) | 1 | - | - | 189 (93.1%) | 102 (79.1%) | 1 | - | - |
| | Yes | 22 (8.8%) | 40 (24.5%) | 3.37 | 1.92–5.93 | <0.001¶ | 6 (18.8%) | 28 (29.8%) | 1.84 | 0.68–4.96 | 0.224 | 14 (6.9%) | 27 (20.9%) | 3.57 | 1.79–7.11 | <0.001¶ |
| Fever | No* | 217 (86.8%) | 89 (54.6%) | 1 | - | - | 25 (78.1%) | 60 (63.8%) | 1 | - | - | 177 (87.2%) | 76 (58.9%) | 1 | - | - |
| | Yes | 33 (13.2%) | 74 (45.4%) | 5.47 | 3.39–8.83 | <0.001¶ | 7 (21.9%) | 34 (36.2%) | 2.02 | 0.79–5.17 | 0.136 | 26 (12.8%) | 53 (41.1%) | 4.74 | 2.76–8.15 | <0.001¶ |
| Myalgias | No* | 189 (75.6%) | 56 (34.4%) | 1 | - | - | 17 (53.1%) | 47 (50.0%) | 1 | - | - | 181 (89.2%) | 84 (65.1%) | 1 | - | - |
| | Yes | 61 (24.4%) | 107 (65.6%) | 5.92 | 3.84–9.13 | <0.001¶ | 15 (46.9%) | 47 (50.0%) | 1.13 | 0.51–2.53 | 0.76 | 22 (10.8%) | 45 (34.9%) | 4.41 | 2.48–7.81 | <0.001¶ |
| Headache | No* | 110 (44.0%) | 31 (19.0%) | 1 | - | - | 14 (43.8%) | 32 (34.0%) | 1 | - | - | 99 (48.8%) | 57 (44.2%) | 1 | - | - |
| | Yes | 140 (56.0%) | 132 (81.0%) | 3.35 | 2.10–5.32 | <0.001¶ | 18 (56.3%) | 62 (66.0%) | 1.51 | 0.67–3.42 | 0.325 | 104 (51.2%) | 72 (55.8%) | 1.20 | 0.78–1.87 | 0.415 |
| Loss of Taste/Smell | No* | 243 (97.2%) | 66 (40.5%) | 1 | - | - | 30 (93.8%) | 46 (48.9%) | 1 | - | - | 199 (98.0%) | 74 (57.4%) | 1 | - | - |
| | Yes | 7 (2.8%) | 97 (59.5%) | 51.02 | 22.61–115.13 | <0.001¶ | 2 (6.3%) | 48 (51.1%) | 15.65 | 3.53–69.27 | <0.001¶ | 4 (2.0%) | 55 (42.6%) | 36.97 | 12.95–105.61 | <0.001¶ |

*Reference category

# Analyses could not be done because some categories had a frequency of 0

¶ Denotes statistical significance. PCR: Polymerase Chain Reaction

**Table 3. Multivariable logistic regression analysis of factors associated with seropositivity in the three occupational risk groups.**

| Variable | Category | HIGH RISK | | | MODERATE RISK | | | LOW RISK | | |
|---|---|---|---|---|---|---|---|---|---|---|
| | | OR | 95% CI | p-value | OR | 95% CI | p-value | OR | 95% CI | p-value |
| Sociodemographic | | | | | | | | | | |
| Sex | Male* | 1 | – | – | 1 | – | – | 1 | – | – |
| | Female | 1.35 | 0.73–2.48 | 0.33 | 1.47 | 0.54–4.01 | 0.44 | 1.05 | 0.59–1.84 | 0.87 |
| Ethnic Group | Non-Mestizo* | 1 | – | – | 1 | – | – | 1 | – | – |
| | Mestizo | 0.92 | 0.34–2.47 | 0.86 | 1.24 | 0.28–5.41 | 0.77 | 0.73 | 0.20–2.65 | 0.73 |
| Age | Mean (SD) | 1.01 | 0.98–1.03 | 0.52 | 1.01 | 0.97–1.03 | 0.67 | 1.02 | 0.98–1.05 | 0.20 |
| History of Symptoms | | | | | | | | | | |
| Cough | No* | 1 | – | – | 1 | – | – | 1 | – | – |
| | Yes | 2.13 | 1.00–4.49 | 0.05 | 1.06 | 0.31–3.55 | 0.92 | 1.73 | 0.72–4.15 | 0.22 |
| Fever | No* | 1 | – | – | 1 | – | – | 1 | – | – |
| | Yes | 1.88 | 0.95–3.69 | 0.06 | 1.92 | 0.63–5.82 | 0.24 | 2.29 | 1.12–4.68 | 0.022¶ |
| Myalgias | No* | 1 | – | – | 1 | – | – | 1 | – | – |
| | Yes | 2.07 | 1.12–3.83 | 0.021¶ | 0.53 | 0.19–1.46 | 0.22 | 1.56 | 0.71–3.42 | 0.26 |
| Headache | No* | 1 | – | – | 1 | – | – | 1 | – | – |
| | Yes | 1.15 | 0.61–2.19 | 0.66 | 1.09 | 0.41–2.96 | 0.85 | 0.68 | 0.38–1.21 | 0.19 |
| Loss of Taste/Smell | No* | 1 | – | – | 1 | – | – | 1 | – | – |
| | Yes | 30.8 | 13.20–71.86 | <0.001¶ | 17.71 | 3.77–83.17 | <0.001¶ | 25.03 | 8.45–74.14 | <0.001¶ |

*Reference category.

¶ Denotes statistical significance

students/teachers) during a sampling period from January to September 2021, we found that seropositivity was 39.5% (CI 95% 34.9–44.4) for the HR group, 74.6% (CI 95% 66.4–81.4) for the MR group, and 39.0% (CI 95% 34.0–44.4) for the LR group.

It is important to interpret our data in the context of community transmission and SARS-CoV-2 attack rates to understand why we obtained high seropositivity amongst all groups, and in particular amongst the MR group. In Ecuador, surveillance of SARS-CoV-2 infections was carried out and reported by the Public Ministry of Health (Ecuadorian Public Ministry of Health COVID-19 Reporting Website): during the sampling period of our study (January to September, 2021), a peak in daily COVID-19 confirmed cases (2,085 new cases) was reported on April 27th, representing a 2.5-fold increase when compared to the 834 new cases reported on January 13th (the beginning of our sampling) [36, 37]. The HR group was sampled during a period of low daily COVID-19 transmission with sampling ending before the April 2021 peak. Thereafter, new daily cases remained above 1,000 until mid-July when they started to decline reaching 219 new cases on September 27th (our last sampling date) [36, 37]. When looking into SARS-CoV-2 attack rates in occupational risk groups in Quito, several studies using Real-Time Quantitative Reverse Transcription Polymerase Chain Reaction (RT-qPCR) have reported infection rates of 20.7%, 12.3%, and 15.2%, amongst funeral home workers, police officers, and food delivery riders, respectively [38–40]. No differences in infection rates were seen according to age and sex [38–40]. Other studies, looking at community transmission in other regions of Ecuador have reported variable infection rates ranging from 6.74% to 49% [41–44]. A study by Morales-Jadan et al., using RT-qPCR in four, mainly Andean, provinces of Ecuador (Bolivar, Chimborazo, Tungurahua, and Napo) observed an overall PCR positivity of 26.2% (95%CI, 23.6%-29%); however, when looking into the individual provinces, Tungurahua Province had the highest infection rate with 65.3% (95%CI, 58.7–71.4%) [41]. This finding is relevant for our study given that Ambato, where our MR group was recruited, is the

provincial capital of Tungurahua; this might also explain the high seropositivity (74.6% [CI 95% 66.4–81.4]) we found in this group. Another study by Rodriguez-Paredes et al., done in the coastal province of Manabí, reported an overall RT-qPCR positivity of 16.13%; however, when looking into individual communities they found infection rates as high as 35% in El Carmen and 33% in Pedernales [42]. All of the aforementioned studies were done between April and September, 2020, 9–12 months prior to our sampling period; however, these studies indicate that infection rates in Ecuador were high amongst occupational groups as well as communities in the Andean, Coastal and Amazonian regions, resulting in extensive community transmission before our sampling period. Our high seropositivity results in all occupational exposure groups reflect the high SARS-CoV-2 attack rates and community transmission in Ecuador from the start of the pandemic.

When looking into other seropositivity studies in Latin America, over a similar sampling period, we found 57 articles (from Argentina, Bolivia, Brazil, Chile, Colombia, Dominican Republic, El Salvador, Mexico, Nicaragua, Peru, and Venezuela) representing a total of 123,277 participants; the lowest seropositivity reported was 7.2% in the city of Divinopolis, Brazil and the highest was 85.0% (95%CI, 82.1–88.0) in the Dominican Republic [45].

With respect to studies done in Andean countries over the same time period, 7 studies with a total of 4,515 participants [23, 25, 46–50], reported IgG seropositivity ranging from 28% [50] to 59% [46]; the majority with seropositivity below 53% [25, 47–50], consistent with our estimates for HR (39.5%) and LR (39.0%) groups. Previous seropositivity studies done in Ecuador include several studies from the coastal region of Ecuador, and largely from a single community, Atahualpa, in Santa Elena Province [12–17, 19, 51], and only one study done in the Andean region on the country in the city of Cuenca [18]. In these studies, the estimated seropositivity ranged from 11.7% (95%CI, 10.0–13.6) to 63.2% (95%CI, 57.4–68.6) [12–19, 51]. The pre-vaccination seropositivity observed here, despite sampling having done approximately 1 year after these other studies, remained within this seropositivity range; further, none of these previous studies evaluated potential occupational risks.

Front-line professionals, including but not limited to doctors, nurses, hospital workers, and police officers, may be at an increased risk of SARS-CoV-2 infection, and might be expected to have greater seropositivity than the general population [29, 52, 53]. A systematic review of healthcare workers prior to vaccination in the United States, Europe and East Asia report a seropositivity ranging from 0.3% to 32.6% with a pooled seropositivity of 8.5% (95%CI,7.1–9.9); additionally, the study detected greater seropositivity in men and in older HCWs (>40 years) in Europe and East Asia, while in the United States seropositivity was greater in younger HCWs [54]. Only two previous studies have reported seropositivity for HCWs in the Andean region: a study of 1,021 Colombian HCWs reported post-vaccination seropositivity of 35% (95% CI, 31.7–38.6) based on detection of IgG to the SARS-CoV-2 nucleocapsid antigen [48]; and a study 783 Bolivian hospital staff (n = 783) before vaccination reported a seropositivity of 43.4% (95%CI, 38.8–48.0) [25], similar to that reported for the HR group of HCWs in our study. Neither of these studies evaluated the effects on seropositivity of direct patient contact (i.e., higher risk of SARS-CoV-2 exposure) [25, 48]–here, we observed a 3-fold greater risk of seropositivity among HCWs with direct patient contact compared to those with minimal contact.

Our MR group consisted of individuals with occupations that did not allow them to self-isolate due to the nature of their work (e.g., merchants, farmers, and those involved in the transportation of livestock). A previous study done in the Andean region, over a similar sampling period, analyzed a population of agricultural workers linked to dairy, livestock/meat and milk and poultry in Colombia [47], and observed a seropositivity of 39.2%; seropositivity was associated with low educational level, urban residence, living in more crowded households,

and travel between districts [47]. Here, we observed a much higher seropositivity (74.6% for MR group) during the same sampling period in the Andean region representing the second highest reported in Latin America [45]; a possible explanation for such high seropositivity may relate to the high mobility of this risk group that included merchants and farmers who were constantly engaged in inter-municipal and inter-provincial travel to live-stock fairs to sell their product. In Ecuador, each district was responsible for the formulation and implementation biosafety guidelines in their respective populations during the pandemic, such that implementation varied between districts; in Ambato, strict measures were implemented to reduce transmission including fines (equivalent to 10% of the minimum monthly salary) for not wearing a mask, and prohibitions on the of selling of goods in the streets (a common practice in Ecuador and in Ambato) [55]. The latter would have represented an important loss of income for a significant proportion of the population engaged in informal commerce. Temporary migrations of informal merchants to other districts with less severe restrictions or less effective enforcement could have led to a higher risk of SARS-CoV-2 infection. Additionally, as discussed previously, a study done in June 2020, reported a high RT-qPCR positivity of 65.3% (95%CI, 58.7–71.4%) in Tungurahua, the province in which Ambato is the provincial capital [41]. On the other hand, the higher seropositivity in the MR group in comparison to our two other groups might also be explained by later sampling compared to the other occupational groups, although there was some overlap with the LR group.

In our study, the symptoms consistently associated with seropositivity across risk groups were a past history of loss of taste and/or smell. Other symptoms were also associated with seropositivity, but varied between groups. COVID-19 can present with various symptoms and several studies have investigated the relationship between COVID-19 seropositivity and these symptoms. A previous multicentric study carried out in more than 40 countries by the Global Consortium for Chemosensory Research reported that 1,468 out of 3,386 participants diagnosed with COVID-19 suffered from loss of both smell and taste at the beginning of their illness [56]. Furthermore, a prospective study in Pennsylvania, USA by Cao et al. observed that self-reported loss of smell and taste were associated with greater than 30-fold odds of SARS-CoV-2 seropositivity [57]. Here, we observed that a history of loss of taste and/or smell was associated with greater seropositivity of IgG antibodies, irrespective of occupational risk group. With respect to other symptoms, two studies by Mesenburg et al. and Terças et al. in Brazil reported that the most commonly presented symptoms associated with SARS-CoV-2 seropositivity, were alterations in smell/taste, palpitations, fever, tremor or chills, and difficulty breathing [58, 59]. The 2nd Brazilian study, EPICOVID, included data from 77,075 individuals and associated seropositivity with symptoms and the presence of a chronic non-communicable disease (NCD). EPICOIVOID observed that cough, myalgias, chills, and difficulty breathing were significantly associated with seropositivity among those with an NCD [58]. In this study, presence of significant associations of seropositivity with specific symptoms (e.g., fever and myalgia) varied between groups and might be explained by differences between groups in socio-demographic factors such as age or rates of asymptomatic or unreported infections.

Limitations. This study is subject to several potential limitations. Selection bias was minimized by close coordination with leaders, stakeholders and institutional authorities; however, convenience sampling within each occupational study group was used which may limit the generalizability of our results. Misclassification biases were minimized by use of a standardized data collection instrument (ISARIC WHO) by a trained research team. Underreporting of SARS-CoV-2 exposures were likely because of limited access to PCR-based diagnostics, especially during the early months of the pandemic in Ecuador; this may have resulted in systematic errors. The hospital-based group would be expected to have the lowest level of underreporting of confirmed infections. Given the significant impact of COVID-19 symptoms

in the context of the pandemic, participants would have been more likely to accurately report such symptoms, although recall bias cannot be excluded and many of the symptoms are shared with other infectious diseases present in Ecuador. In this study, we refer to seropositivity rather than seroprevalence. The two concepts are often used interchangeably in the literature; however, seroprevalence can be a measure of past exposure to an infection which results in long-lasting or life-long immunity [60]. Given the fast dynamics of COVID-19 disease and the uncertainty associated with its corresponding immune response, seropositivity mis a more appropriate term to be used for measuring the proportions of people with relatively recent serological evidence of exposure.

## Conclusions

COVID-19 seropositivity was significantly associated with a history of loss of taste and/or smell regardless of presumed occupational exposure risk. Healthcare workers with direct contact with patients had higher seropositivity compared to those with minimal contact. Seropositivity was greatest in the moderate risk group, presumably because of a high degree of mobility for work and limited use of or access to protective equipment.

## Supporting information

**S1 File. ISARIC-WHO Global COVID-19 Clinical Platform Questionnaire.**
(DOCX)

## Acknowledgments

Seroprevalence ECU-Group: Tatiana Veloz[1], Mónica Pérez[1], María José Montero[2], Grace Noroña[2], Santiago Jácome[2], Luis Jiménez[2], Jessica Ruiz[2], Nicole Pérez[2], Mayra Bonito[2], Evelyn Benavides[2], Jomayra Escobar[2], Jennifer Caya[2]

 1. Universidad Internacional del Ecuador, Medical School, Quito, Ecuador
 2. Universidad Tecnica de Ambato, Medical School, Ambato, Ecuador
 Lead Author: Natalia Romero-Sandoval; nromero@uide.edu.ec

## Author Contributions

**Conceptualization:** Jose E. Leon-Rojas, Miguel Martin, Irina Chis Ster, Philip Cooper, Natalia Romero-Sandoval.

**Data curation:** Jose E. Leon-Rojas, Fernanda Arias-Erazo, Miguel Martin, Irina Chis Ster, Philip Cooper, Natalia Romero-Sandoval.

**Formal analysis:** Jose E. Leon-Rojas, Miguel Martin, Irina Chis Ster, Natalia Romero-Sandoval.

**Funding acquisition:** Jose E. Leon-Rojas, Patricia Jiménez-Arias, Ricardo Recalde-Navarrete, Philip Cooper, Natalia Romero-Sandoval.

**Investigation:** Jose E. Leon-Rojas, Fernanda Arias-Erazo, Patricia Jiménez-Arias, Ricardo Recalde-Navarrete, Angel Guevara, Josefina Coloma, Miguel Martin, Irina Chis Ster, Philip Cooper, Natalia Romero-Sandoval.

**Methodology:** Jose E. Leon-Rojas, Angel Guevara, Josefina Coloma, Miguel Martin, Irina Chis Ster, Philip Cooper, Natalia Romero-Sandoval.

**Project administration:** Jose E. Leon-Rojas.

**Resources:** Patricia Jiménez-Arias, Ricardo Recalde-Navarrete, Angel Guevara, Josefina Coloma.

**Supervision:** Miguel Martin, Irina Chis Ster, Philip Cooper, Natalia Romero-Sandoval.

**Validation:** Fernanda Arias-Erazo, Miguel Martin, Irina Chis Ster, Philip Cooper.

**Writing – original draft:** Jose E. Leon-Rojas, Fernanda Arias-Erazo, Patricia Jiménez-Arias, Ricardo Recalde-Navarrete, Angel Guevara, Josefina Coloma, Miguel Martin, Irina Chis Ster, Philip Cooper, Natalia Romero-Sandoval.

**Writing – review & editing:** Jose E. Leon-Rojas, Fernanda Arias-Erazo, Patricia Jiménez-Arias, Ricardo Recalde-Navarrete, Angel Guevara, Josefina Coloma, Miguel Martin, Irina Chis Ster, Philip Cooper, Natalia Romero-Sandoval.

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
