## [Decision Letter · Decision Letter 0]

15 Nov 2023

PONE-D-23-24305COVID-19 IgG Seropositivity and its determinants in occupational groups of varying infection risks in two Andean cities of Ecuador before mass vaccinationPLOS ONE

Dear Dr. Leon-Rojas,

Thank you for submitting your manuscript to PLOS ONE. After careful consideration, we feel that it has merit but does not fully meet PLOS ONE’s publication criteria as it currently stands. Therefore, we invite you to submit a revised version of the manuscript that addresses the points raised during the review process.

We look forward to receiving your revised manuscript.

Kind regards,

Mariana Lourenço Freire, Ph.D

Academic Editor

PLOS ONE

Journal Requirements:

2. Please include a complete copy of PLOS’ questionnaire on inclusivity in global research in your revised manuscript. Our policy for research in this area aims to improve transparency in the reporting of research performed outside of researchers’ own country or community. The policy applies to researchers who have travelled to a different country to conduct research, research with Indigenous populations or their lands, and research on cultural artefacts. The questionnaire can also be requested at the journal’s discretion for any other submissions, even if these conditions are not met.  

Please find more information on the policy and a link to download a blank copy of the questionnaire here: https://journals.plos.org/plosone/s/best-practices-in-research-reporting. 

Please upload a completed version of your questionnaire as Supporting Information when you resubmit your manuscript.

3. Please expand the acronym “CEDIA” (as indicated in your financial disclosure) so that it states the name of your funders in full.

5. One of the noted authors is a group or consortium: Seroprevalence ECU-Group

In addition to naming the author group, please list the individual authors and affiliations within this group in the acknowledgments section of your manuscript. Please also indicate clearly a lead author for this group along with a contact email address.

Reviewers' comments:

Reviewer's Responses to Questions

**Comments to the Author**

1. Is the manuscript technically sound, and do the data support the conclusions?

Reviewer #1: Partly

Reviewer #2: Yes

Reviewer #3: Yes

2. Has the statistical analysis been performed appropriately and rigorously? 

Reviewer #1: I Don't Know

Reviewer #2: Yes

Reviewer #3: Yes

3. Have the authors made all data underlying the findings in their manuscript fully available?

Reviewer #1: Yes

Reviewer #2: Yes

Reviewer #3: Yes

4. Is the manuscript presented in an intelligible fashion and written in standard English?

Reviewer #1: Yes

Reviewer #2: Yes

Reviewer #3: Yes

5. Review Comments to the Author

Reviewer #1: This manuscript describes COVID-19 IgG Seropositivity and factors contributing to seropositivity in occupational groups of varying infection risks in Ecuador before mass vaccination. The authors have compared three different occupational risk groups based on an expected level of exposure to SARS-CoV-2: high, median, and low risk (HR, MR and LR, respectively). Several factors including specific symptoms were studied to investigate if they were associated with being seropositive for antibodies against SARS-CoV-2.

I have several questions and also think that the authors should have focused more on the age (and sex) difference between the groups, regions and time of sampling in the discussion.

I wonder why the group expected to have median exposure risk had significantly higher seropositivity than the other two groups. Why did the high-risk group have such a low seropositivity? Was it because they were mostly health care workers using personal protective equipment (PPE)? This could have been discussed further, se more below.

Does Ecuador have a registry of infection based on confirmed COVID-19? This would have been interesting to know. Also, if such a register exists, did it show a difference between different regions of Ecuador? (as this manuscript focuses on two cities).

Abstract

Where is the number of infections and deaths reported in the first line of the abstract reported and when (reference)? These numbers are not repeated and referenced in the introduction. The numbers also seem a little high. According to worldometer on the 30th of August 2023, the numbers for South America was 69 million cases and 1.4 million deaths.

In the objectives, the part about studying determinants potentially associated with seropositivity is not mentioned.

In the conclusion, I would suggest using another word than “significant” to describe the proportions as this term if commonly used when discussing statistical significance.

Introduction

lines 38-43 is one long sentence and the meaning is difficult to understand. I suggest that you rephrase.

Lines 48-52, a bit detailed with all the different world regions, it is perhaps better to focus more on South/Latin America.

Line 58-59 states that it is limited information on seropositivity in occupational groups and associated factors. I guess you mean in Ecuador as line 59 and 56 have many references studying this.

Materials and methods

I wonder if it is a “problem” that the participants came from different regions of Ecuador? What is known about the spread of the pandemic between different regions and time points?

In line 85-86 it states that the recruitment in the HR group started in January 2021 and that the LR and MR groups in July 2021, but how long did it the sampling continue for the different groups? What was the range of the sampling time? If the LR and MR groups were sampled 6 months later, could that explain why there were more seropositive samples among LR than HR? Update: The recruitment dates are given in Table 1. The HR group was sampled between January and February 2021 and the MR group was sampled between 21 July and 27 September. This is unfortunate. How did the pandemic develop between January and September 2021 in Ecuador?

Line 118-119: the numbers of participants in each group add up to 882, not 873.

The LR group was much younger than the other two groups. Did you try to age-match the participants? Younger individuals could be less worried of getting COVID-19 and behave differently than older individuals. According to Table 1, there was also a sex difference between the groups with HR and MR being 66.4% and 61.4%, respectively, but the LR groups was only 37.2% women.

The individuals in each group should ideally have been more similar in age, samples collected in the same region and at the same time point to optimize the comparison between the groups.

Was the access to PCR-testing similar in the different regions? Or was it easier access to testing in the HR group?

Table 1: in the HR risk, 38.4% reported to have been positive. This is very close the number of positive samples in this group (39.9%). I think it is surprising that there has not been more undetected disease in this group. Usually, seropositive numbers are higher than reported cases. Is it because this group was tested very frequently?

Table 1: The MR group had a shorter time from COVID-19 diagnosis than the other two groups (95% CI did not overlap for the HR and MR groups). Could this have influenced the seropositivity with waning of antibodies?

Table 1- age: I don’t understand the Age category. Is it the SD or the range of age that is shown? Is age given only for positive cases? I think it should be given for all participants in the group with the mean/median and range. And possible between infected/non-infected if there was an age difference between these groups.

Results

Line 144. It seems that the MR group was significantly different from the other two groups based on the 95% CI.

Line 151, a questionnaire is mentioned with 20 symptoms. Is this questionnaire added supplementary data? There are only 5 symptoms included in table 2. How very these 5 chosen from the 20 in the questionnaire?

Lines 160-165 mentions a subgroup analysis of the HR group between those with direct patient contact versus minimal patient contact. How many participants were included (overall and in the two groups)? This finding is also mentioned in the conclusion, lines 277-278 and more information about these sub-analyses should be included.

Table 3 – was age a continuous variable? I don’t understand the entry.

Table 2 and 3: There is a line dividing the myalgia row that is not in the other symptom categories. I think this line should be removed.

Discussion

First paragraph: are there any other seroprevalence studies from Ecuador?

Line 185: The 44.6% estimate (overall?) is not mentioned in results.

Reference 38 in line 203, from Colombia, measured seroprevalence using antibodies against the nucleocapsid and not spike/RBD. I found it strange that seropositivity was only 35% after completion of vaccination as most vaccines at that time were based on spike/RBD. I would rephrase this sentence.

Lines 221-231: about the restrictions in different districts of Ecuador. What were the differences in restrictions between the regions included here?

I think the authors could have speculated further on why the seropositivity of the MR group was so high, such as different sampling periods etc.

Line 143: levels of IgG in relation to loss of taste and smell. These levels are not mentioned or shown in results only positive/negative results.

Lines 251-254: I don’t understand what the authors want to say here. The LR group had several symptoms associated with seropositivity, but this was the youngest group. Less severe infections have been more common in younger individuals. The groups with the highest seroprevalence and less positive test, was the MR group (suggesting higher asymptomatic infections. But asymptomatic infections also could give lower levels of antibodies and less seroconversion). Could it be that the MR group was too small to detect associations?

Reviewer #2: The work in the present manuscript by Leon-Rojas et al. addresses the COVID-19 IgG Seroposittivity and its determinats in occupational groups of varying infection risks in two Andean cities of Ecuador before mass vaccination. Thus, this manuscript addresses an urgent medical problem – The presence of anti-SARS-CoV-2 IgG antibodies that can be a protector effect against COVID-19 . The idea, implementation, presentation of results and discussion are convincingly presented. However, a few minor things need to be addressed:

1. Materials and Methods: The study is well-done but suffers lack of information on immune responses after immunization. Since immune responses were not evaluated after vaccination and boost, the following sentences should be revised: “analyse changes in antibody levels over a period one year following vaccination with several COVID-19 vaccines”.

2. Please, in line 78, clarify “ICUs” acronym.

3. Please, in line 107, substitute the “Analyses were conducted on each of three samples” by “Analyses were conducted on each of the samples from three diferent groups”.

4. line 151-154: Whats the difference between symptoms described in your questionnaire and the individual COVID-19 symptomatology? Please, Clarify.

5. Please, to all tables, show the significant difference found with a sign and describe it the table footer.

6. Please, write the questionnaire used in the study during recruitment period and post it as supplementary material.

Reviewer #3: This in an interesting and valuable study about the epidemiology of COVID-19 in Ecuador that reports a high seropositivity values in the population after 1 year and a half of COVID-19 pandemic and prior to the introduction of vaccines.

However, there are some major concerns that the authors should address to improve the quality of the manuscript:

1. The period of analysis is from January to September 2021. Is is possible to show the values for each risk group and for the whole month for each month of the study, or at least bi-monthly? My impression is that community transmission was happening from the beginning of the pandemic so perhaps as early as January 2021, the authors will find the same high seropositivity values.

2. These results endorsed that SARS-CoV-2 community transmission was happening in Ecuador since early stages of the pandemic. This is why no matter the risk group considered, the values are really high. I am missing important references and a wider discussion of this topic in this manuscript. There are several articles done in Ecuador during the first semester of COVID-19 pandemic that even using RT-PCR testing, a non cummulative approach like serology to address prevalence of infection, were able to detect really high attack rates:

- Interestingly, three of this manuscript refers to risk occupational groups like police, food delivery riders or funeral home workers (See the DOI for this papers: DOI: 10.3389/fpubh.2022.1012434; DOI: 10.3389/fmed.2021.735821; DOI: 10.1016/j.scitotenv.2021.145225). This papers should be discussed in the context of this manuscript as clearly endorse the same findings that the authors get on this study: a high prevalence of SARS-CoV-2 infection on occupational risk groups.

- Also, few more manuscripts also endorse extended community transmission in general population even in rural and remote locations in Ecuador. These studied should be also discussed in the context of this manuscript as they endorse the findings of the authors: even in low risk group the prevalence is really high. This is expected if those results are put on the context of the findins of these mentioned articles that show extensive community transmission all over Ecuador (See the DOI to revise and cite: DOI: 10.3389/fmed.2023.1001679; DOI: 10.4269/ajtmh.21-0582; DOI: 10.2471/BLT.20.283028; DOI: 10.22605/RRH7643).

3. The authors should provide more details about the clinical performance of the serological test used. It is important to know the sensitivity of this test as this would impact the true values of seropositivity. For instance, the authors cite papers done in communities from Santa Elena and Esmeraldas, where rapid antigen test were used. That means a lost of sensitivity compared to a good ELISA test and probably an underestimation of prevalence of easily 20% or more. I would expect that the ELISA used in this study has a good sensitivity but no matter that the authors should address the estimated true values considering the sensitivity (and also specificity) of the text used.

4. Those results are quite worrying and means that COVID-19 pandemic was really out of control in Ecuador. I am missing an strong discussion about this, contextualizing this results with the mortality in Ecuador associated to COVID-19 (there are also several papers addressing this topic in Ecuador that the authors should consider to cite). Morevover, the scenario described for Ecuador was happening in other countries in Europe or Asia much more later and basically after omicron wave. It is important to discuss this topic as that means that COVID-19 pandemic would have been poorly managed in Ecuador.

6. PLOS authors have the option to publish the peer review history of their article (what does this mean?). If published, this will include your full peer review and any attached files.

Reviewer #1: No

Reviewer #2: **Yes: **Bernardes, WPOS

Reviewer #3: **Yes: **Miguel Angel Garcia Bereguiain

---

## [Author Response · Author response to Decision Letter 0]

9 May 2024

Dear Editor, 

We would like to thank you for your consideration for publication of our manuscript in PLOS ONE. We would also like to thank you for the opportunity to reply to the reviewers’ comments, requests, and constructive criticisms. 

We have addressed them all and we have also added the suggested, interesting references and altered the manuscript in light of their useful remarks. We now believe that the manuscript, the communication, and the scientific messages have greatly improved as result of all these reflections. Therefore, we very much hope that the manuscript is ready for publication. We have attached both clean and altered manuscripts with tracked changes in red, as requested.

The manuscript now fits PLOS ONE’s style requirements and we have included the questionnaire on inclusivity in global research. The acronym of CEDIA has been expanded in the acknowledgements section and reads: “Corporación Ecuatoriana para el Desarrollo de la Investigación y Academia” (CEDIA). In the same section, we have included the information for the Seroprevalence ECU Group as well as the lead author. Finally, regarding data availability, a simplified database can be accessed openly through: https://bit.ly/4aa1Fdf. 

Dear Reviewers,

Thank you for your insightful comments and constructive suggestions. Below, please find a point-by-point response addressing your remarks.

Reviewer #1

Comment 1: I have several questions and also think that the authors should have focused more on the age (and sex) difference between the groups, regions and time of sampling in the discussion.

Thank you for your insightful comment. 

Serological data collection around the time of the pandemic had enormous difficulties. Matching by any variable would have added another layer of complexity which we thought was unrealistic at the time.

Therefore, our study has not been designed to be comparative. There are three very different occupational groups with a priory envisaged different characteristics and levels of exposure. We aimed at representing them and investigate their seropositivity. For instance, the university population was expected to be younger and healthier than other two, the hospital workers were envisaged to experience higher levels of exposure but also protection, whilst the latter group assumed to have had the average population levels of exposure. Therefore, we think that comparative seropositivity analysis would be misleading. Averaging across the three population would also encounter difficulties in defining the “population denominator”.

Comment 2: I wonder why the group expected to have median exposure risk had significantly higher seropositivity than the other two groups. Why did the high-risk group have such a low seropositivity? Was it because they were mostly health care workers using personal protective equipment (PPE)? This could have been discussed further, se more below.

Thank you for your insightful questions. We found the difference between the MR and HR groups very interesting as well and perhaps we were not discussing it enough. We have improved the discussion related to this beginning at line 279 that now reads: “In Ecuador, surveillance of SARS-CoV-2 infections was carried out by the Public Ministry of Health and reported in official websites and bulletins; from these, during the sampling period of our study (January to September, 2021), a peak in daily COVID-19 confirmed cases (2,085 new cases) was reported in the 27th of April, representing a 2.5-fold increase when compared to the 834 new cases reported in the 13th of January (the beginning of our sampling) [62, 63]. Therefore, our HR group was sampled during a period of low daily COVID-19 transmission and its sampling ended before the aforementioned peak”. Also, starting at line 294: “A study by Morales-Jadan et al., using RT-qPCR in four, mostly Andean, provinces of Ecuador (Bolivar, Chimborazo, Tungurahua, and Napo) observed an overall PCR positivity of 26.2% (95%CI, 23.6%-29%); however, when looking into the individual provinces, Tungurahua had the highest infection rate with 65.3% (95%CI, 58.7-71.4%) [58]. This finding is relevant for our study given that Ambato, where our MR group was recruited, is the capital city of Tungurahua; this might also explain the high seropositivity (74.6% [CI 95% 66.4-81.4]) we found in this group.” And finally, beginning at line 410: “Additionally, as mentioned before, a study done in June 2020, reported a high RT-qPCR positivity of 65.3% (95%CI, 58.7-71.4%) in Tungurahua, the province where Ambato is the capital city. On the other hand, the higher seropositivity in the MR group in comparison to our two other groups might also be due to the sampling period (i.e., it was the last one to be recruited), although there was some overlap with the LR group [1,33].” 

Comment 3: Does Ecuador have a registry of infection based on confirmed COVID-19? This would have been interesting to know. Also, if such a register exists, did it show a difference between different regions of Ecuador? (as this manuscript focuses on two cities).

Yes, we have provided the information regarding the registry as well as the behavior of the epidemic during our sampling period in the discussion section starting at line 279 that reads: “In Ecuador, surveillance of SARS-CoV-2 infections was carried out by the Public Ministry of Health and reported in official websites and bulletins; from these, during the sampling period of our study (January to September, 2021), a peak in daily COVID-19 confirmed cases (2,085 new cases) was reported in the 27th of April, representing a 2.5-fold increase when compared to the 834 new cases reported in the 13th of January (the beginning of our sampling) [62, 63]. Therefore, our HR group was sampled during a period of low daily COVID-19 transmission and its sampling ended before the aforementioned peak. Afterwards, new daily cases stayed above 1,000 until the middle of July when they started to decrease steadily, reaching 219 new cases in the 27th of September (our last sampling date) [62, 63].”

Comment 4: Abstract. Where is the number of infections and deaths reported in the first line of the abstract reported and when (reference)? These numbers are not repeated and referenced in the introduction. The numbers also seem a little high. According to worldometer on the 30th of August 2023, the numbers for South America was 69 million cases and 1.4 million deaths.

Thank you for indicating this to us, we accept that more clarity is required. The referenced number was for Latin America, not South America. Latin America includes Mexico and other countries that are not part of South America. We have changed the numbers to reflect South America only, as requested and have provided the reference in the introduction starting in line 33 that now reads: “COVID-19, caused by the severe acute respiratory syndrome coronavirus 2 (SARS-CoV-2), emerged as a global pandemic estimated at 68.7 million infections and 1.35 million deaths in South America up to February 2024 [1].”

Comment 5: In the objectives, the part about studying determinants potentially associated with seropositivity is not mentioned.

Thank you for pointing this out. We have added this in the abstract starting at line 5 and now reads: “Objective: To estimate SARS-CoV-2 seropositivity and its determinants before vaccination in occupational groups of adults presumed to have different levels of exposure and associations with potential symptomatology.”

Comment 6: In the conclusion, I would suggest using another word than “significant” to describe the proportions as this term if commonly used when discussing statistical significance.

We agree with your assessment, we have changed the word to “Notable”, in line 22 of the abstract.

Comment 7: Introduction. Lines 38-43 is one long sentence and the meaning is difficult to understand. I suggest that you rephrase.

Thank you, we have rephrased it to improve clarity starting at line 42 and now reads: “Numerous studies have reported SARS-CoV-2 seropositivity in different populations worldwide prior to the implementation of vaccination campaigns [7–19]. However, there are relatively few studies from the South American region assessing pre-vaccination seropositivity in occupational risk groups [14, 20].”

Comment 8: Lines 48-52, a bit detailed with all the different world regions, it is perhaps better to focus more on South/Latin America.

We have focused more on Latin/South America, starting at line 50 and now reads: “Another meta-analysis of 88 studies from 34 countries, with sampling dates up to November 2020, estimated a pooled seropositivity of 8% (95% CI 6-10%) in the Americas that was highest in Colombia (29%, 95% CI 23-31%) [2]. Analysis of nine studies done in South America between April and September 2020 estimated a pooled seropositivity of 33.6% (95% CI 28.6-38.5%), [52] with a particularly high seropositivity of 73% observed in indigenous populations [10].”

Comment 9: Line 58-59 states that it is limited information on seropositivity in occupational groups and associated factors. I guess you mean in Ecuador as line 59 and 56 have many references studying this.

 Yes, indeed, we meant in Ecuador, we have improved the clarity of this sentence starting at line 88 and that now reads: “There are limited published information on pre-vaccination COVID-19 seropositivity in occupational groups of varying infection risks and associated factors in Ecuador [12–19].” 

Comment 10: Materials and methods. I wonder if it is a “problem” that the participants came from different regions of Ecuador? What is known about the spread of the pandemic between different regions and time points?

Thank you for your comment. Our objective was to assess different occupational groups and we wanted to include a group composed mainly of merchants/farmers, that was found in Huachi Grande, Ambato. A similar group could not have been possible to find in Quito that, as the capital of the country, has a more heterogenous population. Regarding the spread of the pandemic, within and between regions, there is limited published data in Ecuador. We have included more information of the community transmission and reported infection information of Ecuador during our sampling periods in the discussion starting at line 277 and reading: “It's important to interpret our data in the context of community transmission and SARS-CoV-2 attack rates in order to understand why we obtained high seropositivity amongst all groups, and in particular amongst the MR group. In Ecuador, surveillance of SARS-CoV-2 infections was carried out by the Public Ministry of Health and reported in official websites and bulletins; from these, during the sampling period of our study (January to September, 2021), a peak in daily COVID-19 confirmed cases (2,085 new cases) was reported in the 27th of April, representing a 2.5-fold increase when compared to the 834 new cases reported in the 13th of January (the beginning of our sampling) [62, 63]. Therefore, our HR group was sampled during a period of low daily COVID-19 transmission and its sampling ended before the aforementioned peak. Afterwards, new daily cases stayed above 1,000 until the middle of July when they started to decrease steadily, reaching 219 new cases in the 27th of September (our last sampling date) [62, 63]. When looking into SARS-CoV-2 attack rates in occupational risk groups in Quito, several studies using Real-Time Quantitative Reverse Transcription Polymerase Chain Reaction (RT-qPCR) have reported infection rates of 20.7%, 12.3%, and 15.2%, amongst funeral home workers, police officers, and food delivery riders, respectively [55-57]. No differences in infection rates were seen according to age and sex [55-57]. Other studies, looking at community transmission in other regions of Ecuador have reported variable infection rates ranging from 6.74% to 49% [58-61]. A study by Morales-Jadan et al., using RT-qPCR in four, mostly Andean, provinces of Ecuador (Bolivar, Chimborazo, Tungurahua, and Napo) observed an overall PCR positivity of 26.2% (95%CI, 23.6%-29%); however, when looking into the individual provinces, Tungurahua had the highest infection rate with 65.3% (95%CI, 58.7-71.4%) [58]. This finding is relevant for our study given that Ambato, where our MR group was recruited, is the capital city of Tungurahua; this might also explain the high seropositivity (74.6% [CI 95% 66.4-81.4]) we found in this group. Another study by Rodriguez-Paredes et al., done in the coastal province of Manabí, reported an overall RT-qPCR positivity of 16.13%; however, when looking into individual communities they found infection rates as high as 35% in El Carmen and 33% in Pedernales [59]. All of the aforementioned studies were done from April to September, 2020, 9-12 months prior to our sampling period; however, these studies help to showcase that infection rates in Ecuador were high amongst occupational groups as well as communities in the Andean, Coastal and Amazonian regions, resulting in extensive community transmission since 2020. Therefore, our high seropositivity results, in all occupational exposure groups, respond to this phenomenon of high SARS-CoV-2 attack rates, high community transmission and poor management by the government at the beginning of the epidemic in Ecuador.”

Comment 11: In line 85-86 it states that the recruitment in the HR group started in January 2021 and that the LR and MR groups in July 2021, but how long did it the sampling continue for the different groups? What was the range of the sampling time? If the LR and MR groups were sampled 6 months later, could that explain why there were more seropositive samples among LR than HR? Update: The recruitment dates are given in Table 1. The HR group was sampled between January and February 2021 and the MR group was sampled between 21 July and 27 September. This is unfortunate. How did the pandemic develop between January and September 2021 in Ecuador?

Between January and February, 2021 (recruitment period of the HR group), a total of 105,724 new COVID-19 cases were reported in Ecuador. More information regarding the development of the pandemic during our sampling period has been added to the discussion section starting at line 279 and now reads: “In Ecuador, surveillance of SARS-CoV-2 infections was carried out by the Public Ministry of Health and reported in official websites and bulletins; from these, during the sampling period of our study (January to September, 2021), a peak in daily COVID-19 confirmed cases (2,085 new cases) was reported in the 27th of April, representing a 2.5-fold increase when compared to the 834 new cases reported in the 13th of January (the beginning of our sampling) [62, 63]. Therefore, our HR group was sampled during a period of low daily COVID-19 transmission and its sampling ended before the aforementioned peak. Afterwards, new daily cases stayed above 1,000 until the middle of July when they started to decrease steadily, reaching 219 new cases in the 27th of September (our last sampling date) [62, 63].”

Comment 12: Line 118-119: the numbers of participants in each group add up to 882, not 873.

Thank you for pointing this out, it has been corrected, starting at line 173 and now reads: “Of 882 participants included in the analysis, 422, 127, and 333 were in HR, MR, and LR groups, respectively.”

Comment 13: The LR group was much younger than the other two groups. Did you try to age-match the participants? Younger individuals could be less worried of getting COVID-19 and behave differently than older individuals. 

We did not try to age-match the participants because we wanted to analyze the prevalence in three different occupational exposure risk groups. As one of the groups was a university, we expected to have a younger population. We understand that this is a limitation in our study which is why we have analyzed the determinants within each group rather than directly comparing HR vs MR vs LR, as this will lead to biased results.

Comment 14: According to Table 1, there was also a sex difference between the groups with HR and MR being 66.4% and 61.4%, respectively, but the LR groups was only 37.2% women. The individuals in each 

---

## [Decision Letter · Decision Letter 1]

2 Jul 2024

PONE-D-23-24305R1COVID-19 IgG Seropositivity and its determinants in occupational groups of varying infection risks in two Andean cities of Ecuador before mass vaccinationPLOS ONE

Dear Dr. Leon-Rojas,

Thank you for submitting your manuscript to PLOS ONE. After careful consideration, we feel that it has merit but does not fully meet PLOS ONE’s publication criteria as it currently stands. Therefore, we invite you to submit a revised version of the manuscript that addresses the points raised during the review process.

We look forward to receiving your revised manuscript.

Kind regards,

Mariana Lourenço Freire, Ph.D

Academic Editor

PLOS ONE

Journal Requirements:

Reviewers' comments:

Reviewer's Responses to Questions

**Comments to the Author**

1. If the authors have adequately addressed your comments raised in a previous round of review and you feel that this manuscript is now acceptable for publication, you may indicate that here to bypass the “Comments to the Author” section, enter your conflict of interest statement in the “Confidential to Editor” section, and submit your "Accept" recommendation.

Reviewer #1: All comments have been addressed

Reviewer #3: All comments have been addressed

2. Is the manuscript technically sound, and do the data support the conclusions?

Reviewer #1: Yes

Reviewer #3: Yes

3. Has the statistical analysis been performed appropriately and rigorously? 

Reviewer #1: I Don't Know

Reviewer #3: Yes

4. Have the authors made all data underlying the findings in their manuscript fully available?

Reviewer #1: Yes

Reviewer #3: Yes

5. Is the manuscript presented in an intelligible fashion and written in standard English?

Reviewer #1: Yes

Reviewer #3: Yes

6. Review Comments to the Author

Reviewer #1: I am satisfied with the authors’ answers to my questions/comments and the updated manuscript. I support that this manuscript now should be accepted.

Reviewer #3: The authors have improved the manuscript with a better contextualization of their results.

However, there is one of my comments that I believe it was not fully understood. I ask the authors to provide the values for seropositivity for each month of the study, or bi-monthly at least. This is possible to do no matter if only one sample was collected for each individual.

My idea with this comment is to see if there was any trend in seropositivity of if those high values found by the authors were already happening as early as January 2021, what would mean that COVID-19 was spreading massively along 2020 in the Ecuadorian population.

I believe this analysis will make the manuscript even more interesting.

It is up to the authors to include this final recommendation in the final version.

7. PLOS authors have the option to publish the peer review history of their article (what does this mean?). If published, this will include your full peer review and any attached files.

Reviewer #1: No

Reviewer #3: **Yes: **Miguel Angel Garcia-Bereguiain

---

## [Author Response · Author response to Decision Letter 1]

30 Jul 2024

Dear Editor, 

We would like to thank you for your consideration for publication of our manuscript in PLOS ONE. We would also like to thank you for the opportunity to reply to the reviewers’ comments, requests, and constructive criticisms. 

We have addressed them all bellow point-by-point and have highlighted the changes in the manuscript’s text. We have also checked the accuracy of all the included references. We believe that our article has now fulfilled all requirements to be considered for publication in PLOS ONE.

Dear Reviewers,

Thank you for your insightful comments and constructive suggestions. Below, please find a point-by-point response addressing your remarks.

Reviewer #1

Comment 1: I am satisfied with the authors’ answers to my questions/comments and the updated manuscript. I support that this manuscript now should be accepted.

Thank you for your recommendation, your suggestions have certainly improved the clarity of our manuscript.

Reviewer #3

Comment 1: The authors have improved the manuscript with a better contextualization of their results. However, there is one of my comments that I believe it was not fully understood. I ask the authors to provide the values for seropositivity for each month of the study, or bi-monthly at least. This is possible to do no matter if only one sample was collected for each individual. My idea with this comment is to see if there was any trend in seropositivity of if those high values found by the authors were already happening as early as January 2021, what would mean that COVID-19 was spreading massively along 2020 in the Ecuadorian population. I believe this analysis will make the manuscript even more interesting. It is up to the authors to include this final recommendation in the final version.

Thank you for your comment. To improve clarity, we have provided the recruitment dates for each of the three risk groups in the text of Methods. Recruitment of the 3 groups was sequential starting with the HR group and was done between January and September. Seropositivity by month of recruitment is shown in the graph below. As we can see the data are insufficient to allow evaluation of a trend in the transmission of SARS-CoV-2 in these baseline data.

---

## [Editor Report · Decision Letter 2]

13 Aug 2024

COVID-19 IgG Seropositivity and its determinants in occupational groups of varying infection risks in two Andean cities of Ecuador before mass vaccination

PONE-D-23-24305R2

Dear Dr. Leon-Rojas,

We’re pleased to inform you that your manuscript has been judged scientifically suitable for publication and will be formally accepted for publication once it meets all outstanding technical requirements.

Kind regards,

Mariana Lourenço Freire, Ph.D

Academic Editor

PLOS ONE

---

## [Editor Report · Acceptance letter]

19 Aug 2024

PONE-D-23-24305R2 

PLOS ONE

Dear Dr. Leon-Rojas, 

I'm pleased to inform you that your manuscript has been deemed suitable for publication in PLOS ONE. Congratulations! Your manuscript is now being handed over to our production team.

Kind regards, 

on behalf of

Dr. Mariana Lourenço Freire 

Academic Editor

PLOS ONE